# "Boys 'Round Here": Masculine Life-Course Narratives in Contemporary Country Music

## Cenate Pruitt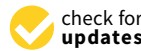

Department of Sociology & Human Services, University of North Georgia, Gainesville, GA 30566, USA; ccpruitt@ung.edu

**Abstract:** Country music remains one of the most popular genres in U.S. American society but is historically under-researched compared to rock, rap and other styles. This article extends the social science literature on the genre by examining themes of masculine identity in popular country hits of the current century. A content analysis of 35 top country hits from the last 15 years of the *Billboard* charts reveals three key masculine archetypes: the lover, the family man and particularly the country boy, which is the dominant masculine image within the last few years of the genre. Together, the three create a life-course narrative where the rambunctious country boy will eventually settle into monogamous heterosexual romance, with marriage and fatherhood presented as the ultimate achievement of successful manhood. A fourth, lesser, archetype, the roughneck, presents an "arrested development" version of the country boy, fully-grown but rejecting the social and familial responsibilities of the other archetypes. These narratives simultaneously challenge some aspects of hegemonic masculinity (urbanity, white-collar labor) while reinforcing others (whiteness, heterosexuality).

**Keywords:** country music; gender; masculinity; life course; fatherhood; identity

## 1. Introduction

Country music, the fourth-most popular radio format in the U.S., comprises 10% of all music consumed in the U.S., whether digital streaming or physical purchases (The Nielsen Company 2017). Long associated with traditional values and small-town life, the genre remains popular in regions outside the Deep South (Shusterman 1999; Bernstein 2016). Rasmussen and Densley (2017) identified the current era of "bro country" as more masculine-oriented and more likely to objectify women than songs from prior decades did. This study operates in parallel to theirs, looking at what country songs from roughly the same timeframe (2000–2015) say about men and male identities and updates Aday and Austin's (2000) longitudinal study of aging imagery in country music for the decade and a half following that publication.

I find that the lyrical portrayals of men and masculine behavior in pop country music from 2000 to 2015 present a specific and clear articulation of country masculinity across the life course: a country man is hard-working, small-town and Christian and reaches true maturity through monogamous heterosexual marriage and raising children. This identity is both exceptionally traditional in its focus on heterosexual relationships, meaningful labor and fatherhood and also works to present rural and exurban men as an ideal rather than as "hicks" or "hillbillies."

## 2. Literature Review

Country music, at its heart, is about the "story-song," melodramatic tales of living, love and loss designed to evoke an emotional response as opposed to a danceable beat or a catchy hook (Aday and

Austin 2000). These songs often present an entire narrative surrounding a life-course event; a broken heart, a new romance, leaving home for the first time and so on (Kurzban 2012; Lewis 1989). These themes are largely universal—everyone can appreciate a song about being in love or a song about having your heart broken. They are universal experiences presented in a specific genre style (Franke 1997). What makes country music country music is the art of exaggeration; this heartbroken man will never love again, this small-town girl is the prettiest one in the world, this country boy works harder and parties harder than any city slicker, a lyrical tradition that can be traced back to the pre-recorded era of tall tales and folk heroes (Peterson 1999; Ching 2001).

## 2.1. Country (Music) Masculinity as a Contested Identity

Country music is sometimes seen as oppositional to the norm, valuing traditionalism and conservative values over the centrist pluralism of the mainstream, with a regressive or even reactionary sense of masculinity (Malone 1993; Shusterman 1999; Bernstein 2016). A counter-claim holds that rural conservative masculinity is still seen as the most authentic masculine ideal in the United States (Campbell et al. 2006; O'Sullivan 2016). This interplay between urban and rural reflects Connell (1995) assertion that the dominant or hegemonic, form of masculinity at any given point in space and time is necessarily a contested one, with the masculine ideals that are valorized in the country genre existing in contrast not only to the feminine ideals presented within the genre but with the masculine images provided by rock, rap and other genres as well. In effect, while country music masculinity may be marginal or oppositional outside of the country music audience, it can simultaneously be the hegemonic ideal within that audience.

For Bertrand (2004), men in country music have always been negotiating an oppositional masculinity. Elvis and his contemporaries were creating a rockabilly identity in postwar America, a white rural ideal in resistance to the "man in the grey flannel suit." The "outlaw" movement of the 70s doubled down on this perspective, standing defiant against both mainstream rock music and the Nashville establishment (King 2014; Waldron 2017). Today's country performers similarly attempt to defy expectations, embracing a traditional-yet-stylized country identity in opposition to the urbane "metrosexual" (Campbell et al. 2006; Rasmussen and Densley 2017). Peterson (1999) and Ching (2001) divide country music, as a genre into "hard" and "soft" forms, with "hard-shell" (Peterson's term) or "hard country" (Ching's term) referring to a specific vision of the country artist as a perpetual outsider, not only to the big city but to the country music establishment itself. The country establishment, then, is the "soft" or "mainstream." While mainstream country is willing to flirt with pop stylings as discussed below, hard country remains true to at least some conception of its roots, continuing the tradition of labeling whatever's going on around it as not *real* country (Peterson 1999; Lacy 2012; Wilson 2015).

## 2.2. Constructing (and Consuming) Country Identities

A significant part of the "country" identity is performed via consumption—most obviously by listening to country music but also of the signifiers of rural life, such as the cowboy hat and the pickup truck. In other words, a college student in the suburbs can perform country masculinity as readily as any farmer or coal miner can, given the right boots, truck and music. This allows men to perform a pseudo-traditional working-class rural manhood regardless of their actual economic status or urban/rural/suburban position (Lacy 2012; O'Sullivan 2016).

Despite the trappings of "regular guy"-ness, the country music establishment itself is upper class and the top-selling artists are multi-millionaires (Peterson 1999). The earliest origins of country music as a recorded genre are largely commercial in nature, as the impresarios who found, recorded and promoted early country stars like Fiddlin' John Carson had no real affection for the genre but had a deep and abiding affection for making money (Peterson 1999). Country music was and remains, a marketable product for the music industry regardless of executives' attitudes towards the music itself. Pecknold (2004) suggested the artists' public image is of working-class mundanity even in the face of economic excess: star country singers perform rurality by owning a pickup truck, a four-wheeler

and other outward indicators, even if the truck is kept in a four-car garage on several acres of private property. What Bertrand identified as a form of resistance, Pecknold saw as a prison.

This performative aspect means the country identity can potentially become a self-parodying "redneck minstrelsy," which Nystrom (2009) and Schwetman (2017) specifically link with campy 1970s fare like *Smokey and the Bandit* and various films starring the actor Joe Don Baker, including *Walking Tall* and *Live and Let Die*, where working-class rural characters are seemingly simultaneously held up for admiration by rural audiences and mockery by more urbane ones. Hauhart (2008) connects this concept to the wildly successful Blue Collar Comedy Tour franchise, particularly Dan "Larry the Cable Guy" Whitney, a Nebraska native who portrays a stereotypical yokel in a Confederate flag baseball cap. Whitney's routine is full of negative images of a variety of groups (immigrants, ethnic minorities, the disabled, gays and lesbians) but by playing a buffoon, Hauhart argues, the audience can convince themselves they are not laughing at the racist joke itself but at "Larry" for being so foolish as to (openly) say such impolite things. Whitney, to his credit, has parlayed the character into a multi-million-dollar industry, bringing in $13 million in 2013 alone (Hayes 2013).

Hubbs (2011) discusses how the cultural conversation around country music intentionally others working-class culture and political values, framing rural whites as the entrenched "bigot class" despite evidence to the contrary. The "redneck" is, to the outside world, a bad image: racist, ignorant, deplorable (O'Sullivan 2016). By consuming songs about "rednecks" and "country folk," Fox (2013) argues country fans often hate the genre and its trappings, constructing positive self-images from the often stereotypical and potentially negative images of rural, blue-collar life on display as a form of resistance, while Malone (1993) claims country lyrics focus on "self-affirming" themes: pride in hard work, simple faith, committed relationships and American patriotism. Hauhart similarly identifies the Blue Collar Comedy Tour audience as not quite rural but rather as suburbanites who idealize rural life and choose to consume these commercialized versions of "blue collar" culture as a rejection of both bourgeois urban values *and* the stereotypically pathologized poverty of J.D. Vance's *Hillbilly Elegy*.

### 2.3. Country Music and "Real America"

Hyden (2014) claims that, since the 1990s, pop-country has largely taken over the musical niche previously occupied by "heartland rock" artists like Bruce Springsteen and John Mellencamp—songs about the triumphs and tribulations of the residents of small towns and rural areas across the United States. Rosen (2013) explicitly describes the sound of Florida-Georgia Line as "closer to John Mellencamp than George Jones." On the one hand, the rise of alternative rock and the decline of the arena rocker opened up an ecological (and economic) niche that pop-country rushed to fill. On the other hand, the big rock artists may have alienated conservative audiences with their personal politics—Mellencamp, a self-identified liberal, has repeatedly challenged Republican politicians for using his songs in their campaign materials, while Springsteen is notably close to the Obamas, performing private concerts for their staff (Janik 2017; Brandie 2017). Furthermore, the "heartland rock" hits of the 80s were decidedly ambivalent in their presentation of rural lives and rural values—neither Mellencamp's "Pink Houses" nor Springsteen's "Thunder Road" would be at home on today's country radio. Pop-country holds no such ambiguity; small towns and rural areas are Sarah Palin's "real America," where people work hard, go to church on Sunday and take pride in themselves.

Country music then secured itself as the new "sound of the heartland," a deeply and intensely patriotic genre in contrast to the jet-setting international sounds of contemporary pop. Coroneos (2013) connects the growth of this quasi-reactionary pop-country to the blackballing of the Dixie Chicks in 2002. The bluegrass-inspired neo-traditionalist sound was on the rise; the Chicks were selling out arenas and the Coen Brothers' *O Brother Where Art Thou?* led to a resurgence of interest in bluegrass, with the film's soundtrack winning both the Country Music Association and the Academy of Country Music's Album of the Year awards and the single "Man of Constant Sorrow" winning a Grammy as well as ACM and CMA awards. However, Dixie Chicks' singer Natalie Maines's statement that "we're ashamed the President of the United States is from Texas" led to a swift and profound

backlash. Griffiths (2015) adds to this analysis by observing that the Chicks' then-current album was not only sonically traditional but *lyrically* traditional, focusing on themes of marriage, motherhood and domesticity, themes that are not inherently conservative but ones that can certainly be embraced by conservative audiences. In the meantime, more "patriotic" fare, often delivered by red-blooded stars like Toby Keith, reaffirmed the status quo of support for the U.S. invasion of Iraq (Boulton 2008; Decker 2019).

However, this may be changing; as Donald Trump ascended to the presidency of the United States in 2017, country music stars began to speak critically on political matters. Country Music Association (CMA) Awards hosts Carrie Underwood and Brad Paisley performed a parody of Underwood's "Before He Cheats" titled "Before He Tweets," targeting Trump's provocative use of social media and Tim McGraw, Faith Hill and Jennifer Nettles of Sugarland all took to social media following mass shootings in Las Vegas and Florida to call for gun control legislation (Resnikoff 2017; Watts and Paulson 2018). Meanwhile, outsider artists like Jason Isbell and Sturgill Simpson actively engaged in political discourse, with Isbell campaigning for Tennessee Democratic candidates and Simpson openly calling Trump a "fascist fucking pig" and his supporters "ignorant fucking bigot[s]" on a social media live stream (Waldron 2017; Zaillian and Bernstein 2018). Wilson (2015) on the other hand, suggests that Simpson et al. represent a sort of palatable country music for the NPR set, appealing to coastal elitists who turn up their nose at NASCAR, Walmart, songs about small towns and patriotism and especially at the people who love all of those things.

## 3. Materials and Methods

The sample was drawn systematically: the top five songs for years ending in 0, 3, 5 and 8 were chosen from the *Billboard* year-end "Hot Country Songs" charts in order to gather a broad overview of lyrical themes in the most popular songs. Ryan et al. (1996) point out the rationale for using *Billboard* charts: songs often enter at a low rank, then climb in popularity before petering out; thus the *Billboard* year-end charts indicate these songs are popular for significant portions of the year in question, with some eventually becoming part of the canon. This method's greatest strength is the sheer ubiquity of the songs chosen; using the biggest hits ensures the songs and messages being discussed reached a large audience. A 2012 decision by *Billboard* to alter the calculations for Hot Country Songs is highly relevant: in addition to country radio, airplay on pop stations—as well as streaming plays and digital sales—now count towards chart position (Levy 2018). On the one hand, this suggests an even stronger degree of influence, as the songs that chart going forward are potentially transcending country radio and breaking out into the mainstream of pop culture; as goes Taylor Swift, so goes country music? On the other hand, the pressure to create hits for pop consumers as well as country fans may lead to a dramatic shift in the nature of the genre. During the submission process for this article, "country trap" artist Lil Nas X's "Old Town Road" achieved the unusual feat of appearing on *Billboard*'s Hot 100, Hot R&B/Rap and Hot Country charts simultaneously, despite being played on a single country radio station in the entire U.S.—the thrust of its success was social media memes and YouTube plays (Kopf 2019). However, as quickly as "Old Town Road" became a hit, it was pulled from the Hot Country Songs charts, as *Billboard* declared it "does not embrace enough elements of today's country music to chart in its current version" (Leight 2019). Another significant disadvantage here is the missed impact of music on the margins; award-winning artists like Isbell, Simpson and Kacey Musgraves receive little-to-no airplay on country radio, having been shuffled off into the realm of "Americana" (aka "Country for People Who Don't Like Country") alongside their outsider forebears Gillian Welch, Steve Earle and Lucinda Williams (Wilson 2015; Waldron 2017; Doyle 2017).

All song lyrics were obtained from Genius.com. In addition to having user-submitted lyrics, Genius offers artists the ability to be verified and post official comments on the lyrics (Linssen 2016; Kehrer 2016). The methodology for this project was a straightforward close line-by-line reading as utilized by Saucier's (1986) study of country lyrics and Weitzer and Kubrin's (2009) analysis of misogynistic content in rap lyrics. In this case, as song lyrics are contingent on understanding the

song's thematic context, the unit of analysis utilized here was "thematic units"; single words or short phrases which can be categorized as an observer, an actor, an action or the target of an action (Weber 1990). For instance, the narrator of "Beer For My Horses" expresses admiration for cowboys of old. The verse in question would contain multiple thematic units–an actor (the narrator) discusses his admiration (an action) for the cowboy heroes (target of an action), while a later line features the cowboys (now actors) enacting vigilante justice (the action) upon criminals (target) and so on.

The initial round of analysis focused on identifying the song's broad themes (broken heart, new romance, nostalgia, etc.) with specific attention paid to identifying the gender of the observers, actors and targets in the song's narrative. A song performed by a man in which the vocalist refers to himself in the first person is then logically about a man unless the lyrics specifically suggest otherwise (which did not occur in this sample), while a song performed by a man or a woman using third-person terms was further investigated for other signifiers such as gendered pronouns or the use of a specifically masculine term like "grandpa." Songs by a woman vocalist which referred to a romantic partner (three of the four songs by women in the sample did so) were assumed to be referring to a heterosexual relationship, with the specific exception of "Girl Crush" by Little Big Town, which is discussed in detail below. In some cases, descriptive terms such as *boss* or *criminal* appear. These terms were evaluated in the lyrical context: a man singing about a blue-collar job is presumably (but not necessarily) singing about a male boss, while other songs used more explicitly gendered terms like "boss man" or "bad boy." Country music dwells in cliché and archetype (Franke 1997; Ching 2001), so when these characters appear, they are treated as archetypal versions.

The next pass looked at how the men in these songs were described, explicitly ("my grand-daddy," "small town southern man") or implicitly (expressing distaste with one's job implies gainful employment; signifiers such as *dirt roads* and *farm towns* imply a rural environment). Earlier research proved inspirational but also limiting here: very few of these songs invoke the classic country archetypes of the heartbroken man, the cowboy or the outlaw, instead of presenting more contemporary takes on tropes that may no longer be relevant. Instead, by looking at how these men were presented as actors and as the targets of action, a slightly different set of archetypes emerged. As a specific example, multiple songs used terms like "father" and "daddy" to describe a protagonist's own father, while terms like "daughter" and "baby girl" serve to establish a character as a father in his own right—here a need for caution is clear, as a term like "girl" could refer to an actual literal infant or a romantic interest or even a target of derision. Indeed, all three of those possibilities are found in this sample; when "How Do You Like Me Now?!" uses the term, it is to sarcastically dismiss the subject of the narrator's ire, "I Saw God Today" refers to the narrator's newborn child and multiple songs including "Cruise" and "Take Your Time" use it to denote a romantic partner.

After establishing a broad conception of who appears in each song, the third round of coding focused further on the actions these characters perform in the narrative. Some actions are physical, such as drinking, working or making love, while others are emotional states best described as 'feelings;' feeling alone, feeling happy and so on. These feelings were most often in reaction to other characters in the narrative, in love with a woman, proud of a father, resigned to the failure of a romance. Another action theme was "desire," as in the desire for alcohol, freedom or romance, with other characters sometimes serving as allies or obstacles towards those goals. Common themes such as a desire to drink alcohol, pride in fatherhood and respect for religion began to emerge at this point, leading to the development of a life-course narrative.

## 4. Results

### 4.1. Masculine Archetypes and the Life Course Narrative

Close-reading the songs revealed three central archetypes of masculine behavior as well as a fourth variation. Four songs specifically feature young, rambunctious men who focus on parties, sexual conquests and alcohol; these I group as *Country Boys*. Three more deal with these rambunctious

men entering heterosexual romantic relationships and beginning to shed their rowdy ways. Half of the songs—seventeen in total—discuss men actively engaged in a committed, monogamous, heterosexual romantic relationship or expressing the desire to be so engaged with a specific person, all of which I identify as *Lovers*. Four of these songs return to the classic country archetype of the heartbroken man (Ching 2001; Pruitt 2006) but are ultimately still about wanting or missing the love of a specific individual. Five songs focus on adult men dealing with the responsibilities of maturity, such as work, marriage and child-rearing, which I categorize as *Family Men*. Another three songs feature adult men who still engage in the rough behavior associated with Country Boys, despite being seemingly old enough to know better: these are the *Hell-Raisers*.[1]

These songs present a clearly articulated life course for men who identify with country music. Young men are expected to be rough, rowdy and rambunctious Country Boys for a time, enjoying their youth with as much vigor and recklessness as possible. This includes conspicuous consumption of alcohol, outdoor partying and pursuing women as sex partners. However, there comes a time when the young man must settle down. Becoming a Lover involves sacrifice for the Country Boy, who must abandon his wild ways in favor of meeting his partner's needs and prioritizing her interests alongside or even over his own ("Crash My Party," Sam Hunt's "House Party"). Finally, marriage, fatherhood and career turn the Country Boy fully into the Family Man. He has matured and developed new responsibilities to his spouse, his children and his community ("Small Town Southern Man," "Love Like Crazy"). A Country Boy who stays reckless, keeps partying and refuses to be tied down ages into the Hell-Raiser ("As Good As I Once Was," "How Do You Like Me Now"). Appendix A features a breakdown of the sample into these categories, plus those which feature a Country Boy transitioning into the Lover role, as well as three songs which do not immediately fit into any of these categories.

Several songs, including "Red Dirt Road," "Small Town Southern Man" and "Love Like Crazy" explicitly depict this life-course narrative. The protagonist of "Red Dirt Road" was wild and reckless, crashing cars and drinking beer but with a combination of heterosexual commitment and Christian faith he is now older and wiser, looking back fondly on his youthful misadventures. In "Small Town Southern Man," the protagonist lacks the youthful wildness of the Country Boy but the narrative is fully centered on the importance of love, Christianity and hard work in the protagonist's life and the success of his idealized masculinity. The title character marries his sweetheart, works hard to provide for his family and finds comfort in his faith and his patriotism until his death, at which point he is explicitly stated to be in heaven. "Love Like Crazy" presents the protagonist as "crazy" for marrying young and following his dreams but also makes clear that these values are admirable, as the protagonist's reckless pursuit of goals results in financial success, marital bliss and the respect of his entire community.

*4.2. The Country Boy*

The cowboy of old (whether a literal cowboy or the metaphorical country singer, truck driver or other working man as cowboy) has largely been replaced in this data set by the Country Boy archetype. Instead of busting broncos and roping steers, this pop-country protagonist parties on the riverbank and drives a tricked-out pickup truck. He stands in firm contrast to the urbane city slicker: the "Boys 'Round Here" prefer Hank Williams, Jr. to The Beatles, which is appropriate considering Williams' "A Country Boy Can Survive" is perhaps the codifier of the archetype. All four of the sample's "core" Country Boy songs make much of the Country Boy's rural identity, refer specifically to farming and

---

[1]   Three songs in the sample dealt with primary themes outside this masculine life-course narrative. In "Something More," a woman discusses her dissatisfaction with her career and a desire for escape. "It's Five O'Clock Somewhere" similarly concerns the narrator's desire to flee the drudgery of work for the pleasures of alcohol, while "19 Somethin" presents a nostalgia-fueled longing for the freedom of youth contrasted against the mundane responsibility of adulthood. Intriguingly, all three of these songs to some degree go against the lionization of hard work found in the rest of the sample, providing a clear direction for further research into modern country music.

small towns and at least passively dismiss city life and city dwellers as weaker than or inferior to country life and country people. "Rain Is a Good Thing" (Bryan) begins by observing that city folks complain when it rains but Country Boys know rain helps their crops grow and creates the ideal conditions for "muddin.'"[2]

The Country Boy exists as a post-industrial commodified version of classic country archetypes. As farm life transitions from riding the range to operating heavy machinery, tales of farmers and cowboys may no longer resonate (U.S. Department of Labor 2017). Instead, Country Boy is one of many consumption-based identity choices for young men to adopt. For those who choose the Country Boy identity (or have it thrust upon them), pop country is the musical lingua franca providing a ready-made set of clichés and archetypes to inform their performance.

As much as a backward baseball cap and baggy jeans are symbolic shorthand for hip-hop, cowboy boots and a pickup truck are consumable symbols for the Country Boy. Conspicuous consumption is part of the Country Boy song sample: Florida Georgia Line's "Cruise" invokes Chevrolet trucks, Southern Comfort liqueur and KC HiLites truck accessories, while "Kick the Dust Up" compares drinking straight from a jar to drinking overpriced drinks in a big-city bar. This brand-conscious country performance creates a degree of authenticity but also an explicit performance of class comparable to that found in hip-hop lyrics (Goldman 2007; Gregorio and Sung 2009; Mohammed-baksh and Callison 2015). While these songs do not reference the sort of luxury brands found in hip-hop (e.g., Bentley, Benz, Gucci), the sort of heavily modified pickups discussed in these songs can easily cost over $50,000. To be this kind of Country Boy clearly requires a financial outlay. This tracks well with Pecknold's assertion that being country does not necessarily mean being poor; consider the Robertson family of *Duck Dynasty* fame, multi-millionaires who embody "country-ness" through their consumption patterns (O'Sullivan 2016). Similarly, one explanation for the Nashville establishment's refusal to embrace Sturgill Simpson is not his politics but his lack of "country" signifiers; he wears t-shirts and sneakers, not rhinestones and cowboy boots (Cross-Smith 2017).

Where Have All the Cowboys Gone?

The Country Boy's sense of place differs significantly from the traditional cowboy archetype, who is a solitary wanderer, riding the range (Malone 1993). Several songs in this sample present country musicians as modern-day cowboys, itself a tradition dating back several decades (Pruitt 2006). These songs all explicitly describe a sense of loneliness and longing for home comparable to classic "lonely cowboy" songs ("Wagon Wheel," "My Front Porch Looking In"). That solitary nature is absent in the Country Boy, who expresses joy and relief at not having to stray far from his rural roots and vehemently dismisses the big city and city-dwellers.

Only two songs feature classical cowboy archetypes; Dixie Chicks' "Cowboy Take Me Away" presents the cowboy as a romantic figure but also as a symbol for an idealized rural lifestyle, while "Beer for My Horses," frames cowboys as mythical Old West figures of law and order. Both songs depict a cowboy who may not exist—the dream lover and the folk hero—and both feature traditionalist artists who wound up on the fringes of country music's establishment: the politically left-leaning Dixie Chicks and the aging outlaw (and noted cannabis enthusiast) Willie Nelson.

*4.3. The Lover*

The Lover is a man wholly dedicated to his romantic relationship. The default protagonist/narrator for country songs, the Lover can appear in a new couple ready to start a life together (Chad Brock's "Yes!," "Just Got Started Loving You"), an established pair serving as aspirational role models ("Love Like Crazy") or the heartbroken lover who either struggles to let go of a failed relationship or

---

2　　"Muddin'" is a rural pastime that involves traveling to remote areas and driving 4 × 4 vehicles into the deepest mud possible without becoming stuck.

desperately seeks reunion with a lost love ("The Man I Want To Be"; Thomas Rhett's "Crash And Burn"). The rambunctious Country Boy becomes the Lover by settling down, as seen in "Red Dirt Road" and "Crash My Party" and sacrificing the homosocial bonds of friendship for a serious relationship.

Good Lovers are loyal, steadfast and prepared to make dramatic life-changes to satisfy their partners. Lover songs are about a performance of passion. Gestures both grand and small dominate these songs as boys give up their rowdy ways and men confirm their commitment and fidelity. Romance is often presented as an escape from mundane, day-to-day life ("What About Now," "Why Don't We Just Dance"). The passion and romance of love allow a man to forget whatever troubles him, put aside the drudgery of work and have a moment of pure enjoyment.

In the past, country songs had a strong focus on the heartbroken fool (Ching 2001), a figure of tragicomic misery epitomized by classic songs such as Hank Williams's "I'm So Lonesome I Could Cry" Williams (1949) and Jim Reeves's "He'll Have To Go" Reeves (1959). This modern sample contains relatively few songs about broken hearts and failed romances. "The Man I Want to Be" and "Crash and Burn" both take a hopeful tack, in which the narrators express a desire to win back their lost loves' hearts. "Crash and Burn" situates its protagonist in the tradition of sad country songs and its lyrics serve as meta-commentary on the genre's tropes, while "The Man I Want to Be" is the narrator's prayer for forgiveness, the opportunity to change and divine intervention to bring back his lost love. In contrast, both "Don't Think I Don't Think About It" and "Wagon Wheel" make it explicit that the relationship is permanently concluded and the protagonist must learn to live with the ramifications of his decisions.

### When Women Sing about Loving Men

Two songs where women vocalists discuss their (intended) lovers, Faith Hill's "The Way You Love Me" and Little Big Town's "Girl Crush," both have a decidedly androcentric approach. The narrator of "The Way You Love Me" discusses at length the pleasure her lover provides but centers her satisfaction as something for him to observe and enjoy. Similarly, while the female narrator of "Girl Crush" describes the lips and body of another woman, this homoeroticism occurs in the context of heterosexual jealousy: the other woman is involved with the man the narrator wishes to possess. In that sense, it serves as a foil for Dolly Parton's "Jolene," Parton (1973) which Hubbs (2015) discusses as an "Other Woman Song," in which the narrator addresses a romantic rival. But while the narrator of "Jolene" describes this Other Woman as superior and begs her for mercy, the narrator of "Girl Crush" wants to possess the Other Woman so she can learn how better to entrap a man. In both cases, these songs center on the man's experience and male pleasure in strikingly voyeuristic terms. A third song, Dixie Chicks' "Cowboy Take Me Away" identifies the classic archetypal image of the cowboy as an idealized dream lover, tied deeply to rurality and open spaces as discussed above.

### 4.4. Family Men

Once a Country Boy has found true love and settled down, he becomes a Family Man; the hell-raising comes to an end and the child-raising begins. This requires the love of a good woman, with heterosexual marriage and paternity presented as the ultimate goal and purpose of life. These songs overwhelmingly focus on daughters, with only "Small Town Southern Man" specifically mentioning sons as a possibility. Parenthood allows these men another direction for emotional openness. In songs about being in love, tears are reserved for the heartbroken but Family Men are allowed to cry with joy at the birth of a child, on a wedding day or on other family-centered occasions.

Family Men are providers and often explicitly discuss working to support the family. "Small Town Southern Man" discusses how the title character engages in rough farm labor to support his family. He may not be financially secure but familial love and Christian faith provide all the support he needs. The Family Man's goal is not only to provide for his wife and children but to raise them right, creating another generation of Lovers, Family Men and the women who love them.

Having settled down, the most excited Family Men get is when there's a particularly good coupon in the Sunday paper ("That's What I Love About Sunday"). A Family Man's success is due not only to his hard work and marital fidelity but also to divine assistance. God is often invoked as the ultimate father figure (George Strait's "I Saw God Today," "Small Town Southern Man"), to whom the Family Man owes gratitude.

The modern family man stands in contrast to classic country songs featuring an absent father or unfaithful husband (cf. Merle Haggard's "Mama Tried," Haggard (1968), Tammy Wynette's "D-I-V-O-R-C-E" Wynette (1968)). Only one song in this sample evokes this archetype: Toby Keith's "How Do You Like Me Now?" which presents a loveless marriage to an absent man as a woman's karmic comeuppance for rejecting Keith's advances in the past. In contrast, work and responsibility may separate Family Men from their families but they consistently express the desire to return home as soon as possible. "My Front Porch Looking In" makes this connection explicit; while the narrator invokes the country-singer-as-wandering-cowboy archetype, he also makes clear that his greatest pleasure is returning home to his family at the end of the tour.

*4.5. Hell-Raisers*

Finally, the Hell-Raiser serves as a developmental fork in the life-course narrative. A country boy who never settles down eventually becomes a Hell-Raiser. In this sample, the Hell-Raiser is exclusively found in songs by Toby Keith, who was in his late thirties at the height of his career in the early 2000s. The Hell-Raiser is a Peter-Pan version of the Country Boy: while he has physically aged, he has not emotionally matured. He continues to be rowdy and reckless well into middle age, drinking, carousing and generally "raising hell." Keith's "As Good As I Once Was" in particular displays a deep insecurity and need to prove oneself against the challenges of old age, presenting tales of fisticuffs and sexual prowess as signifiers of his remaining potency. Hell-Raisers are fun to be around and, compared to the Family Men, lead exciting lives but the schtick can get a little long in the tooth. Consider Keith's last major hit, "Red Solo Cup" Keith (2011), an ode to the ubiquitous party accessory. The video for "Cup" features Keith wandering around a raucous house party, surrounded by dancers and carousers seemingly half his age.

The Hell-Raiser is *not* the dangerous figure found in the prison songs of Marty Robbins, Johnny Cash or Merle Haggard. He will not instigate violence but will readily take up arms for a just cause ("As Good As I Once Was," "Beer for my Horses"). Most importantly, the Hell-Raiser feels no remorse for his actions; he is simply having a good time. The Hell-Raiser is somewhat similar to the personas adopted by hard country artists like Willie Nelson or Waylon Jennings; a man of a certain moral character but rough around the edges and ill-equipped for polite society, city life and the commitment required for a permanent relationship. While Peterson (1999) and Ching (2001) categorize certain performers as "hard" or "outlaw" based upon their musical style and their relation to the Nashville music establishment, the Hell-Raiser is more akin to a stock character and can appear in a song by a mainstream artist or by an outsider. Indeed, every single song in this sample is by definition a mainstream hit and Keith, in particular, is well-integrated into the industry, despite his aggressive image.

## 5. Discussion

*5.1. Recurring Themes of Masculine Identity*

Across the life course proposed above, several concepts appear regardless of whether the subject is a young man asking for a date or a wise old grandfather: Christianity, rural/small-town identity, a performance of average-ness and a homosocial bond.

*5.2. Christianity*

Ten of the songs (28%) present Christian faith as part of country life and thus country manhood. Craig Morgan's "That's What I Love About Sunday" centers its small-town family-life narrative on

church attendance, evoking images of people singing off-key with the choir, hands raised in worship. These are as important as the Sunday paper, backyard football and relaxing naps. Brooks and Dunn's "Red Dirt Road" describes the titular area as a place where he first tried beer but also where he found Christian faith, while the idealized small-town family man of Lee Brice's "Love Like Crazy" advocates for the power of prayer. Several of the love songs present God as being directly and personally responsible for the relationship. Rascal Flatts' "Bless the Broken Road" refers to a string of failed romances as part of a divine plan to unite the narrator with his partner, while Chris Young's "The Man I Want To Be" is the narrator's direct prayer to God to be the man in the title, in hopes of winning back a lost love.

Even the rough and tumble Country Boy is expected to be faithful or to at least do his best. "Boys 'Round Here" and "Red Dirt Road" both suggest whatever sins are committed on Friday night will be forgiven Sunday morning. Overall, to be a man in the country milieu is to be Christian, at least in a broad-strokes believes-and-is-saved sense. Country men are allowed missteps and clearly lack the teetotaling aspect of certain sects but they are open about their faith and know that, however far they fall, forgiveness is possible.

### 5.3. Rural/Small-Town Identity

Approximately fourteen songs (40%) use explicitly rural/small-town signifiers, exemplified by the titles of "Red Dirt Road" and Lonestar's "My Front Porch Looking In." Country manhood is, perhaps obviously, rural or at least exurban. The performance of country masculinity often goes beyond simply rejecting the urban, as in "Boys 'Round Here," which suggests that country folk do not listen to The Beatles but prefer Hank Williams, Jr., specifically referencing his song "A Country Boy Can Survive," Williams (1982) perhaps the ur-text of this sort of masculine identity. In that song, the city is dangerous and violent, doomed to collapse, while "country boys," regardless of where they are in the United States, will survive on their backwoods skill sets. Similarly, Toby Keith's "Beer For My Horses" explicitly positions the city as a violent and dangerous place where crime and corruption are rampant, expressing a desire for violent reprisals against urban criminals. Multiple songs in the sample continue in this tradition where the city is framed as an undesirable space, where people constantly complain (Luke Bryan's "Rain is a Good Thing"), the traffic is atrocious (Sugarland's "Something More") and the drinks are overpriced (Luke Bryan's "Kick the Dust Up"). Country protagonists may have to work in the city but they desperately want to get back to the country or a semi-rural area ("It's Five O'Clock Somewhere," "Red Dirt Road," "Something More").

It is important to point out many of these songs are about "the country" or "small towns," not "the South." By presenting *the concept of rurality* as opposed to a single region as central, country artists, labels and radio stations expand their audience share. One of the biggest country stars of the modern era, Keith Urban is from Australia, a long long way from Nashville. Conversely, rock bands like R.E.M. and rappers like Goodie Mob are decidedly "Southern" in their origins and influences but not quite "country." Indeed, aside from the title, Alan Jackson's "Small Town Southern Man" does little to specifically situate the song in the South, being more concerned with the small-town tropes identified above and the working-class concepts discussed below.

### Whiteness

Whiteness is not necessarily mandatory for the genre; two songs in the sample were performed by Darius Rucker, an African-American. Intriguingly, both of Rucker's hits are exceptionally traditional in topic and tone; "Don't Think I Don't Think About It" Rucker (2008) and "Wagon Wheel" Rucker (2013) are both classic "broken-heart" songs, with "Wagon Wheel" further being a cover of Americana act Old Crow Medicine Show. However, Rucker's race is framed in complicated ways by the music videos for the two songs. The video for "Don't Think I Don't Think About It" (Isham 2009) intersperses footage of Rucker playing and singing by himself in various rural environments (a field, a barn) with a series of narrative vignettes featuring a white, heterosexual, couple enacting the breakup discussed

in the lyrics, rather than presenting Rucker himself as the romantic figure. Similarly, Rucker is the only person of color in the "Wagon Wheel" video (Wright 2013) but his identity as a country artist is seemingly authenticated by cameos from other "country" figures, including members of the band Lady Antebellum and the cast of *Duck Dynasty* (Schillaci 2013).

While Rucker may need to be legitimated by a traditional "country" sound and image, white artists are allowed more freedom to either adopt or reject various signifiers of blackness, specifically hip-hop instrumentation and style. Sam Hunt's "House Party" Hunt (2015) features a drum machine and an explicit lyrical callout to Rock Master Scott & the Dynamic Three's "The Roof Is On Fire" while Florida-Georgia Line's "Cruise" Florida Georgia Line (2012) found success on Top 40 radio with a remix featuring '90s rapper Nelly (Anne-Helene 2013; Jessen 2013). Conversely, "Boys 'Round Here" simultaneously appropriates and rejects hip-hop culture, with vocals delivered in a rap-style manner and the use of samplers and Autotune but lyrics that specifically decry "The Dougie," a hip-hop dance step popularized in Dallas, as something alien to country life. The video for the song tries to have it both ways as well, featuring four African-American men in sunglasses, sports jerseys and Wu-Tang Clan t-shirts cruising in a lowrider alongside singer Blake Shelton's pickup truck. Shelton and the men then party together in a barn, including a scene where a white man in camouflage overalls uses a pair of turntables DJ-style (Fanjoy 2013). However, all of the songs where white men adopt hip-hop stylings are from 2010 or later, performed by artists under the age of 40 at the time of recording, can be considered part of the subgenre identified as "bro-country" (Rosen 2013; Rasmussen and Densley 2017; Hyden 2014) and fit into my typology as being about Country Boys; it seems older, more established artists are less likely to engage in the blurring of lines, sticking to traditional instrumentation and themes.

### 5.4. A Performance of Average-ness

In approximately twenty-five (71%) of these songs, country masculinity is specifically presented as "average," using terms like *ordinary* or referring to oneself as a "guy." The men in these songs, grounded in a blue-collar milieu, express joy at a coupon for ground chuck ("That's What I Love About Sunday") and discuss the adventures to be found behind the wheel of a seven-hundred-dollar car (Lonestar, "What About Now"). These men may not be well-educated but neither are they stupid: they are simply "guys." Even when country stars sing about life on the road ("Home," "My Front Porch Looking In"), it is as an unpleasant complication keeping them away from home rather than as an enjoyable or fulfilling success.

As in these two songs, this sample as a whole features a complicated relationship between men and labor. On the one hand, when work appears in these songs, it is consistently seen as worthwhile but difficult, with farming commonly invoked in the sense of working the land ("Boys Round Here," "Rain Is a Good Thing," "Kick the Dust Up"). Other songs opt for vague images of blue-collar work ("That's What I Love About Sunday"). White-collar labor appears explicitly only once: the protagonist in "Love Like Crazy" does *something* with "computers" and becomes wealthy when he sells his product to Microsoft—but critically, he does not lose the titular small-town values. Interestingly, office work is only specifically mentioned in James Otto's "I Just Got Started Loving You," when the vocalist, a man, tells his partner to stay home for a romantic interlude and suggests her work is less important than their pleasure. Essentially, work is only problematic when it infringes on a man's freedom; such as when the narrator of Alan Jackson's "It's Five O'Clock Somewhere" bemoans his disrespectful boss and a lack of time off[3].

---

[3] "Something More" presents a parallel narrative from a woman's perspective, expressing a similar desire to escape a miserable "boss man" and the banalities of the suburbs, with a particularly interesting gendered tell being the woman vocalist's desire for red wine as opposed to beer/liquor. Further analysis of alcohol use imagery in country music might prove quite fruitful (no pun intended).

*5.5. Homosocial Bonds*

Country music has always featured intimate relationships between men, whether as close friends, brothers or fathers and sons (Hubbs 2015). In this sample, the primary homosocial relationships are either platonic friendships or intergenerational bonds between (grand)fathers and (grand)sons. Four songs feature male artists discussing fathers or grandfathers, largely reverentially as role models and dispensers of wisdom ("Beer For My Horses," "Small Town Southern Man"), although in Mark Wills' "19 Somethin'" the narrator's father maintains his embarrassing 1970s haircut well into the 1980s, a single (gentle) presentation of a father as incompetent or silly. Of the eleven songs featuring male friendships, almost all present barrooms and/or social drinking as a bonding mechanism. Toby Keith's "As Good As I Once Was" features the narrator and his male friend getting into a fight while playing pool, while "Beer for my Horses" calls for a group of male vigilantes to celebrate their actions with whiskey. Luke Bryan's "Crash My Party" expresses the narrator's willingness to abandon his friends for a chance to see his love interest; their male bonding is secondary to his romantic desires.

## 6. Conclusions

The male life-course narrative and its primary characters present a fundamentally conservative world view, centered on faith, family and hard work. Young men are allowed a certain degree of immaturity but must eventually submit to heterosexual marriage, fatherhood and blue-collar labor to be considered "real men," with the alternative being a state of arrested development full of temporary pleasure that leaves no lasting legacy. This expands the current state of the literature on masculinity by introducing an analysis of a specific and popular musical genre, while it also expands the literature on country music by articulating and identifying a popular thematic narrative that thus-far has gone largely undiscussed.

*6.1. Directions for Future Research*

This article is designed to work in parallel with recent research in the field such as Rasmussen and Densley (2017) analysis of changing roles for women in country music. More significantly, if assumptions by Levy (2018) and others about an ongoing shift in country music towards the mainstream of pop music are correct, then this piece serves as a sort of theoretical benchmark; this is not only a time capsule of country on the verge of a massive transformation but a beachhead for future analyses to build upon. As discussed above, the biggest mainstream stars of today are no longer afraid of being politically left of the traditional country audience. Crossover country is on the rise, as *Billboard's* biggest country song of 2018 was a collaboration between bro-country rockers Florida Georgia Line and New York City-born songstress Bebe Rexha. A new generation of "hard country" artists like Isbell and Simpson are winning critical acclaim and playing *Saturday Night Live* while openly rejecting conservative politics. There do seem to be limits, as the announcement that Lil Nas X of "Old Town Road" fame would be partnering with Wrangler on an exclusive line of jeans was met with backlash on Instagram, with the brand's social media component responding to aggrieved consumers with affirmations about celebrating "the cowboy spirit in us all" and "this era of music, culture and style" (Elitou 2019).

This genre shift then clearly has an economic component; if success in country music is now being gauged by airplay on Top 40 radio and streaming sites, then country songwriters must necessarily start tailoring the content to appeal to an audience beyond the Blue Collar Comedy crowd. Hit songs are written to be hits, not to be transgressive, like Peterson, Ching and others have made clear over the years. To cater to a mainstream pop audience, will stars like Carrie Underwood eventually go against conservative dogma on politics? If Hyden and Coroneos are correct and the bellicose patriotism of Toby Keith filled the hole left behind by the vanishing of "heartland rock" and the sidelining of more progressive artists into the "Americana" category, what will happen when today's top stars get "too liberal"? As long as that core conservative audience has purchasing power, someone is going to cater

to them, whether it be Toby Keith, the assortment of country second-stringers and throwback acts that performed at Trump's inauguration or something waiting in the wings on YouTube, where right-wing troupes such as "The Deplorable Choir" perform country tracks like "Proud to be Deplorable" and "Fake News Song" while sporting "Make America Great Again" hats (Casey 2017; Thomas 2018; The Deplorable Choir 2019). Has the culture war come to country music at last? Obviously, that research will need to be carried out in the years to come, built around songs that might not have been written yet.

Another vector for lyric-based exploration would be going into greater depth and detail beyond this admittedly small sample. While Ching (2001) analysis of country music masculinity is extensive, it is also almost twenty years old and focused on "hard" artists like Merle Haggard, George Jones and Buck Owens, while largely eliding exploration of more mainstream artists like Kenny Rogers, Garth Brooks or George Strait. As such, one possibility would be a much deeper dive into the history of country music, reaching as far back as the 1950s, when *Billboard* first began keeping records if not earlier and exploring how masculine roles have changed or if they have changed at all. Another approach would be to expand the sample size of the current era, drawing the top five or ten year-end songs from 2000–2015 to see if this larger data set produces a different result or simply reinforces this initial exploration. Finally, purposive data collection built around identity themes might prove fruitful; focusing on songs performed by the small cohort of African-American country singers from Charlie Pride to Cowboy Troy to Darius Rucker may develop an alternative view of manhood, while a dedicated exploration of songs where women sing about men (whether desirable or undesirable) could provide yet another angle on the expectations country music holds for its audience.

A third approach would be audience-driven, interviewing country music fans of various backgrounds and exploring the extent to which these songs resonate with them. Do they take the lyrics seriously? Do they even listen to the lyrics or just focus on the rhythm and melody? In what ways do they embody these country identities in their day-to-day lives, if at all? Is there a divide when it comes to embracing crossover artists like Taylor Swift and Florida-Georgia Line and if so, what factors influence that division; age, gender, socioeconomic status? Like the metaphorical frontier, the path for future work in this area is wide open.

*6.2. Limitations and Implications*

This is a small sample, drawn in a semi-representative manner. Even a casual follower of country music will note the absence of megastar Taylor Swift in the data set, which is an unintended consequence of the collection strategy. A broader lack of women's voices may be due to the sampling methodology utilized or, arguably, the noted lack of high-profile women in the genre at the moment (Guarino 2016; Wilson 2015). As an example of this dearth of talent, between 2006 and 2018 the Academy of Country Music (ACM) and CMA awards for Female Vocalist of the Year were awarded year after year to either Carrie Underwood or Miranda Lambert, with two exceptions—2009 CMA winner Taylor Swift and 2019 ACM winner Kacey Musgraves. Even though Underwood had three #1 *Billboard* Hot Country Songs hits in 2008, she does not appear on the year-end charts I utilized and thus does not appear in the sample. On the other hand, a prominent country music industry figure recently compared "females" in country music to "tomatoes in the salad," suggesting that the problem may be less with the sampling strategy and more with sexism within the Nashville establishment (Wilson 2015).

Beyond the size of the data set, another limitation of this study was financial. The analysis was carried out by a single researcher operating on a very limited budget. The coding was largely done by going over printed copies of lyrics with a highlighter and a pen and making notes in Google Docs, as opposed to using qualitative software packages like NVIVO. As such, there are certainly concerns to be had with the coding process and the possible outcomes but again, the results I offer here are exploratory and meant to serve as a launching point for further study within the genre. Similarly, the lack of actual lyrics in the body of this article is intentional—securing the appropriate clearances for songs held by just one of the major publishing houses would have cost upwards of $3000 U.S. dollars, pending approval by the actual copyright holders.

Outside of academia, this life-course model has potential implications for a variety of professional fields. First, professionals in the caring fields (social workers, counselors, etc.) may be able to utilize the narrative framework presented in these songs to reach out to clients who are country music enthusiasts. Relating individual struggles to the thematic trials and tribulations of these songs, particularly for men who feel they have failed to live up to these expectations due to deindustrialization, job loss and the automation of agriculture may prove fruitful. Social and political organizing in suburban, exurban and rural areas with large numbers of country music fans could also appeal to these narratives as part of broader campaigns, utilizing the narratives of productive labor, fatherhood and family to court key demographics. A particularly cheeky suggestion would be for country music songwriters themselves to actively engage with these narratives; their presence in hit songs indicates significant resonance with the country music audience and as such, finding new ways to tell the same old story just might lead to a number one hit.

**Funding:** This research received no external funding.

**Acknowledgments:** I wish to thank Alana Berry, Heather Moulton-Meissner, Beth Sherouse, and Kerry Wendt for editing and feedback, Rachel Prevatt for bringing the term "redneck minstrelsy" to my attention, Jim Femino at Songstarters! Inc. for graciously letting me take up his time, the anonymous reviewers who've provided feedback on this piece, and lastly (but not least) Loren Norman, Gabriel Owens, James Rodatus, and Wayne Walton for their regular and inspiring icosahedral narrative resolution skills.

**Conflicts of Interest:** The author declares no conflict of interest.

## Appendix A. Discography, Sorted by Life-Course Status, Year and Chart Position

### *Appendix A.1. Country Boys*

Luke Bryan. "Rain Is a Good Thing." *Doin' My Thing*. Capitol Nashville, 2010. (2010 #2)
Florida Georgia Line. "Cruise." *Here's to the Good Times*. Republic Nashville, 2012. (2013 #1)
Blake Shelton featuring Pistol Annies and Friends. "Boys 'Round Here." *Based on a True Story . . . .* Warner Bros. Nashville, 2013. (2013 #3)
Luke Bryan. "Kick the Dust Up." *Kill the Lights*. Capitol Nashville, 2015. (2015 #4)

### *Country Boys/Lover Transition*

Brooks and Dunn. "Red Dirt Road." *Red Dirt Road*. Arista Nashville, 2003. (2003 #5)
Josh Turner. "All Over Me." *Haywire*. MCA Nashville, 2010. (2010 #4)
Luke Bryan. "Crash My Party." *Crash My Party*. Capitol Nashville, 2013. (2013 #4)
Sam Hunt. "House Party." *Montevallo*. MCA Nashville, 2015. (2015 #3)

### *Appendix A.2. Lovers*

Chad Brock. "Yes!" *Yes!* Warner Bros. Nashville, 2000. (2000 #2)
Lonestar. "What About Now." *Lonely Grill.* BNA, 2000. (2000 #3)
Dixie Chicks. "Cowboy Take Me Away." *Fly*. Monument, 1999. (2000 #4)
Faith Hill. "The Way You Love Me." *Breathe*. Warner Bros. Nashville, 2000. (2000 #5)
Rascal Flatts. "Bless the Broken Road." *Feels Like Today*. Lyric Street, 2004. (2005 #3)
Rascal Flatts. "Fast Cars and Freedom." *Feels Like Today*. Lyric Street, 2004. (2005 #5)
James Otto. "Just Got Started Loving You." *Sunset Man*. Raybaw/Warner Bros. Nashville, 2007. (2008 #1)
Blake Shelton. "Home." *Pure BS*. Warner Bros. Nashville, 2008. (2008 #5)
Josh Turner. "Why Don't We Just Dance." *Haywire*. MCA Nashville, 2009. (2010 #3)
Hunter Hayes. "I Want Crazy." *Hunter Hayes*. Atlantic, 2013. (2013 #5)
Sam Hunt. "Take Your Time." *Montevallo*. MCA Nashville, 2014. (2015 #1)
Little Big Town. "Girl Crush." *Pain Killer*. Capitol Nashville, 2014. (2015 #2)

*Broken Hearted Lovers*

> Darius Rucker. "Don't Think I Don't Think about It." *Learn to Live*. Capitol Nashville, 2008. (2008 #4)
> Chris Young. "The Man I Want to Be." *The Man I Want to Be*. RCA Nashville, 2009. (2010 #5)
> Darius Rucker. "Wagon Wheel." *True Believers*. Capitol Nashville, 2013. (2013 #2)
> Thomas Rhett. "Crash and Burn." *Tangled Up*. Valory Music Group, 2015. (2015 #5)

*Appendix A.3. Family Men*

> Lonestar. "My Front Porch Looking In.." *From There to Here: Greatest Hits*. BNA, 2003. (2003 #1)
> Craig Morgan. "That's What I Love about Sunday.." *My Kind of Livin.'* Broken Bow, 2004. (2005 #1)
> George Strait. "I Saw God Today." *Troubadour*. MCA Nashville, 2008. (2008 #2)
> Alan Jackson. "Small Town Southern Man." *Good Time*. Arista Nashville, 2007. (2008 #3)
> Lee Brice. "Love like Crazy." *Love like Crazy*. Curb, 2009. (2010 #1)

*Appendix A.4. Hell-Raisers*

> Toby Keith. "How Do You Like Me Now?!" *How Do You Like Me Now?!* (2000 #1)
> Toby Keith, featuring Willie Nelson. "Beer for My Horses." *Unleashed*. DreamWorks Records, 2003. (2003 #2)
> Toby Keith. "As Good As I Once Was." *Honkytonk University*. DreamWorks Nashville, 2005. (2005 #2)

*Appendix A.5. Miscellaneous*

> Mark Wills. "19 Somethin'." *Greatest Hits*. Mercury Nashville, 2002. (2003 #3)
> Alan Jackson and Jimmy Buffett. "It's Five O'Clock Somewhere." *Greatest Hits Volume II*. Arista Nashville, 2003. (2003 #4)
> Sugarland. "Something More." *Twice the Speed of Life*. Mercury Nashville, 2005. (2005 #4)

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
