# Peer review of "“Boys ‘Round Here”: Masculine Life-Course Narratives in Contemporary Country Music"

_socsci, doi:10.3390/socsci8060176_

Round 1

Reviewer 1 Report

This paper is well-written, and presented well-researched literatures in music and masculinity and thus it seems fitting for a piece of exploratory research that outlines the manhood present in the country songs that sit atop the country music charts. In fact, throughout the sections my only question emerging from the literature is why the author(s) chose the idea of the “Hell raiser” instead of utilizing the popular idea of the outlaw or the “hard country” concept in the literature.  I would also like to applaud the author for engaging the reader without citing lyrics, which is nearly impossible anymore with tightening copyright laws.

Still, this is a social science journal rather than a literary journal, so I would not be comfortable with publication without some discussion of the study weaknesses in conclusion. The biggest issue is the incredibly small sample size of songs, which the methods do not describe as analyzed in any methodical way like the grounded theory method or some type of coding assess with a reliability rating. Also, there simply must be some explicit mention of arguments following the field of literature sparked by Richard Peterson which argues pop country presents a fabricated version of rural manhood by “artists” who look like runway models in designer jeans and cowboy hats. Lastly, further directions of research could be pointed out like larger sample sizes assess with a word count software (see Eastman and Pettijohn) explorations of Sturgill Simpson as an outlier or looking more into how women sing of men on the country charts.

Reviewer 2 Report

This is a very well-written, readable and engaging article.  The arguments and ideas flow very nicely, and the paper is logically organized.  There are some especially compelling observations in the paper: I very much like how the author used the concept of "Redneck Minstrelsy" (it might be worth considering someone like comedian Daniel Whitney's "Larry the Cable Guy" character, with respect to this concept; there is also the obvious corollary in rock of what I'd call working-class minstrelsy, a la Bon Jovi and especially Bruce Springsteen--glamorous multi-millionaire rock icons singing romantic-empathetic songs on MTV about dockworkers, unemployed steelworkers, etc).  The notion of the Hellraiser character type as the "Peter Pan" version of the Country Boy type is likewise clever and compelling. I have virtually no disagreement's with the observations in the paper: they seem very sound to me, for the most part.   Overall, I think the author is correct in asserting that country music, relative to other popular music genres, is under-researched, and so this paper would be a welcome contribution to country music scholarship.  

Here's my primary concern: who is surprised to learn that country music comprises a conservative narrative of masculinity?  The content analysis, it seems to me, reveals something we already know, though it presents it in a nicely-package and coherent form: country songs are largely concerned with hetero-normative tropes and narrative structures.  The conclusion is very unsatisfying.  I was left asking "So?"  I would like for there to be more critical analysis here, rather than mere observation.  What accounts for the prevalence of this narrative?  Why is it so important for country music to rigorously re-inscribe traditionalist notions of masculinity, family, rural life, etc?  What are some of the broader socio-cultural/socio-political implications (gender, sexuality, class,etc.), since this a "social science" journal.  

I do like this article a lot, I should reiterate.  As a musicologist and cultural historian, though, this sort of scholarship gets up my nose a bit.  The content analysis is totally legitimate; but tying it to some musical analysis would, I think, add some interest and depth to this study.  There are musical cliches at play in these songs that very strongly reinforce the narrative structure/life course and character types that are revealed by the content analysis.  Perhaps such an analysis is not possible at this time, but that's where I think a study like this could go in the future.  

Again, my primary concern is the conclusion: there's a lot of data offered, and the case for what is going on in these songs--and in the genre more generally--is very clearly made.  But the conclusion is so brief and unsatisfying!  This study feels cut-off, as the very point when I would like to know more, and would like to see the author explore the implications of the research in greater detail and depth.  

Reviewer 3 Report

Eminently publishable. Just a few minor errors -- misuse of semi-colon LN 10; missing word Ln 46; LN 95, why an end quote?

Author Response

Eminently publishable. Just a few minor errors -- misuse of semi-colon LN 10; missing word Ln 46; LN 95, why an end quote?

The errors you've observed have been corrected - the semi-colon replaced with a connecting word, the missing word has been added, and the end quote was an artifact from an earlier draft and has been removed.

Thank you!